# DiffuseLoco: Real-Time Legged Locomotion Control with Diffusion from Offline Datasets

**Xiaoyu Huang**[*,1]**, Yufeng Chi**[*,1]**, Ruofeng Wang**[*,1]**, Zhongyu Li**[1]**,**
**Xue Bin Peng**[2]**, Sophia Shao**[1]**, Borivoje Nikolic**[1]**, Koushil Sreenath**[1]
[1] UC Berkeley, [2] Simon Fraser University [*]

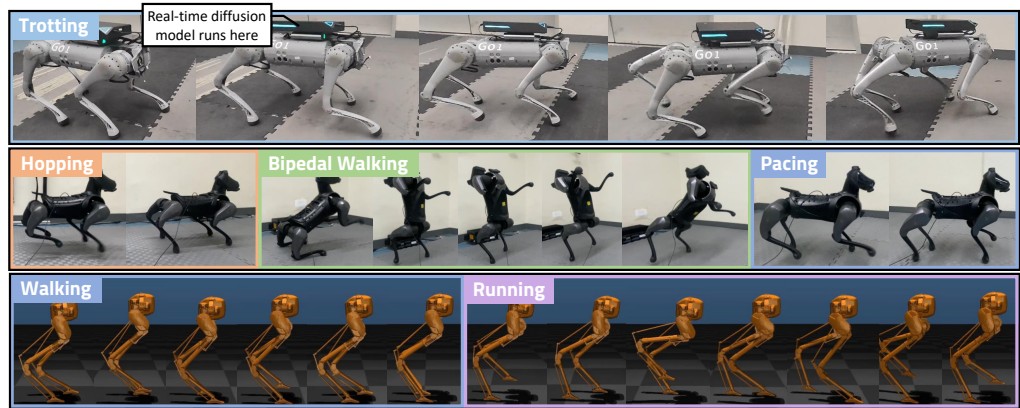

Figure 1. Snapshots of (top) a quadrupedal robot trotting with DiffuseLoco control policy via onboard computing; (middle) DiffuseLoco policy performing a sequence of challenging skills, including hopping, bipedal locomotion, and pacing with smooth skill transitioning; (bottom) a bipedal robot with DiffuseLoco transitioning from walking to running stably. We encourage the readers to watch supplementary videos on these runs.

**Abstract:** Offline learning at scale has led to breakthroughs in computer vision, natural language processing, and robotic manipulation domains. However, scaling up learning for legged robot locomotion, especially with multiple skills in a single policy, presents significant challenges for prior online reinforcement learning (RL) methods. To address this challenge, we propose DiffuseLoco, a novel, scalable framework that leverages diffusion models to directly learn from offline multimodal datasets with a diverse set of locomotion skills. With design choices tailored for real-time control in dynamical systems, including receding horizon control and delayed inputs, DiffuseLoco is capable of reproducing multimodality in performing various locomotion skills, zero-shot transferred to real quadruped robots and deployed on edge computes. Through extensive real-world benchmarking, DiffuseLoco exhibits better stability and velocity tracking performance compared to prior RL and non-diffusion-based behavior cloning baselines. This work opens new possibilities for scaling up learning-based legged locomotion control through the scaling of large, expressive models and diverse offline datasets. For more details, please visit project website `https://diffuselo.co`.

**Keywords:** Offline Learning, Bipedal Walking, Imitation Learning

## 1 Introduction

Learning from large-scale offline datasets has led to breakthroughs in a large variety of domains, such as computer vision and natural language processing, where scaling up both the size of models and datasets leads to improved performance and generalization [1, 2]. This has led to the devel-

---

[*]Equal contribution. Correspondence to Xiaoyu Huang x.h@berkeley.edu

8th Conference on Robot Learning (CoRL 2024), Munich, Germany.

opment of powerful generative models, like diffusion models, which are able to model complex multi-modal data distributions [3, 4] and generate high-quality images and videos.

In robotics, learning from diverse offline datasets has also been shown to be an effective and scalable approach for developing more versatile policies for domains such as robotic manipulation [5, 6] and autonomous driving [7, 8, 9]. However, these domains typically involve agents that have low-dimensional action spaces (*e.g.*, end-effector trajectory) with low re-planning frequency on inherently stable systems (*e.g.*, robot arms or cars). For dynamical systems featuring higher degrees of freedom and more complex dynamics, such as legged robots, data-driven approaches have largely been focused on online reinforcement learning (RL) techniques [10, 11, 12]. Unlike offline learning, it can be difficult to scale online RL to both large models and datasets due to the requirements of online rollouts. Most prior works involving online RL have focused on dynamic motions with a small number of skills for each policy, and scaling towards a single policy that can reproduce a diverse set of challenging locomotion skills remains an open problem.

To address the challenges of offline learning from diverse data sources and learning a set of diverse skills, we present *DiffuseLoco*, a framework leveraging expressive diffusion models to learn multi-modalities in diverse offline datasets without manual skill labeling. Once trained, our controllers execute robust locomotion skills on real-world legged robots for real-time control.

The primary contributions of this work include a novel, versatile framework that learns directly from a diverse offline dataset, demonstrating the benefits of scalable offline learning for practical locomotion skills, and a state-of-the-art multi-skill controller that learns both agile bipedal walking and quadrupedal locomotion skills within a single policy and is deployable zero-shot on real-world quadrupedal robots. Extensive real-world validation shows higher stability and lower velocity tracking errors compared to baselines, with smooth skill transitioning and robustness on varying terrains. This work demonstrates the feasibility of zero-shot learning a diverse locomotion policy directly from a static dataset, opening new possibilities for large-scale learning for real-time, low-level control for complex dynamical systems.

## 2 Related Work

### 2.1 Multi-skill Reinforcement Learning in Locomotion

Recent works in model-free RL have demonstrated promising results in agile locomotion skills for real-world legged robots [12, 13, 14, 15, 16, 17]. Prior works have shown impressive performances on agile skills like jumping, running, and sharp turning on bipedal robots [18, 19], and walking on two feet with quadrupedal robots [20, 21, 22], requiring high agility and robustness. However, these skills are trained with single-skill RL and do not scale with more skills.

A natural idea of learning multi-skill locomotion is to train skill-specific policies, and then coordinate through high-level planning [23, 24, 25, 26]. However, due to the coordination difficulty, these methods remain unscalable to an increasing number of skills. In comparison, learning multi-skill policies from scratch typically involves parameterized motions [27, 28] with limited applicable motions, and motion imitation methods through either reward shaping [29, 30, 31] or adversarial imitation learning [32, 33, 34, 35]. However, this approach still faces challenges such as the limited expressiveness of simple models in online RL frameworks in learning diverse skills.

In general, learning a diverse, agile policy with online RL remains challenging. For example, while existing RL methods have combined skills like jumping and trotting [29, 35], trotting and standing on hind legs with wheels (without walking) [36], a combination of more diverse and agile skills such as stable bipedal walking, pacing, and hopping in a single policy has not yet been demonstrated.

### 2.2 Offline Learning in Locomotion Control

Compared to online learning, offline learning offers better scalability, a simpler training scheme, and an effective way to re-use data, yet prior works in learning low-level locomotion control from offline

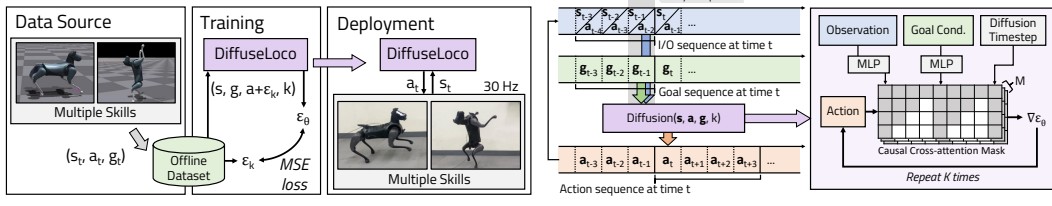

| (a) Workflow Diagram | (b) System Diagram |

Figure 2: (a) The three stages of the DiffuseLoco pipeline: (1) Generate or use an offline dataset with diverse skills gathered by different methods (left); (2) Train DiffuseLoco policy with DDPM loss on dataset trajectories (middle); (3) Deploy on robot hardware and perform agile skills in the real world (right). (b) The DiffuseLoco architecture. At time step $t$, it takes in a delayed $h$-step history of proprioceptive states, goals, and actions. After $K$ denoising iterations, it generates $n$-step future actions and feeds the executed action back to the input.

datasets remain limited. Most prior works focus on simple simulated tasks, such as Gym locomotion tasks, with behavior cloning (BC) [37] and offline RL [38]. Among them, some leverage Q-learning on offline datasets [39, 40, 41], or supervised learning techniques [42, 43, 44, 45]. However, these tasks are oversimplified and do not adequately consider the complexities in real-world scenarios.

An alternative is the use of offline data as a foundation for online learning [46, 47]. Among them, Smith et al. develops baseline policies from offline datasets to bootstrap online learning on real robots. Yet, this approach still requires online learning. Another recent work develops offline learning on humanoid locomotion with real robots [48], with the scope limited to only one walking skill. In comparison, the efficacy of learning completely from offline datasets, especially at a larger scale than a few simple skills, remains unproven in legged locomotion control.

## 2.3 Diffusion Models in Robotics

Emerging efforts to apply diffusion models to real-world robots include using diffusion to manage a variety of manipulation tasks with visual inputs and incorporating self-supervised learning and language conditioning [49, 50, 51, 52]. Yoneda et al. leverages the reverse diffusion process for shared autonomy with a human user in end-effector planning. Additionally, hierarchical frameworks are being developed to handle tasks requiring multiple skills, pushing towards generalist policies [54, 55, 56]. However, these prior works primarily focus on high-level planning on manipulation systems, featuring a low-dimensional action space (*e.g.*, end-effector position), low-replanning frequency (*e.g.*, around 10 Hz), and inherently more stable dynamics.

In contrast, using diffusion models for high-frequency, low-level control remains limited [57]. The most relevant work uses online RL to train diffusion-based actor policies in simpler simulation settings [58], but transitioning to hardware with high-frequency feedback control presents significant challenges due to the instability and rapid dynamics of legged robots [59]. The key difference of this work is demonstrating the applicability of diffusion models in high-frequency, low-level legged locomotion control and the advantages of multimodality and scalability in the real world.

## 3  Versatile Framework for Diffusing Locomotion From Offline Dataset

In this section, we introduce *DiffuseLoco*, a framework for generating and utilizing offline datasets to train scalable multi-skill locomotion policies. DiffuseLoco uses diffusion models to train a low-level multi-skill locomotion policy from datasets with diverse agile skills. A schematic illustration is shown in Figure 2a. Our framework consists of three stages:

**Data Source**  We start by collecting or using an existing offline dataset of multiple skills. To generate the data, we first obtain single-skill control policies, called source policies, conditioned

on goals $\mathbf{g}$ (e.g., velocity commands and base heights). The *source* policies can be trained using different methods, and while we assume the same control frequency, their observation and action spaces can still *vary significantly*. Thus, we instead collect a set of *source-agnostic* state-action-goal pairs across all source policies. For legged robots, the widely-used source-agnostic states and actions are proprioceptive feedback $\mathbf{s}_t$ from the robot and joint-level PD targets $\mathbf{a}_t$. We only collect successful episodes. Although we use simulation data here, DiffuseLoco efficiently re-uses data and can scale to more expensive real-world data collection. More details are in Appendix F.

**Training**  In the second step, we train our DiffuseLoco policy from the offline dataset in an end-to-end manner. Let input state and goal history length be $h$ and output action prediction length be $n$. During training, we sample a segment of state trajectory $\mathbf{s}_{\text{traj}}$ and corresponding action and goal sequences, $\mathbf{a}_{\text{traj}}$ and $\mathbf{g}_{\text{traj}}$. We sample a diffusion step $k$ randomly from $\{1, \ldots, K\}$, and sample a Gaussian noise $\epsilon_k$ to add to the action sequence. Then, a transformer-based denoising model takes the noisy action sequence along with states trajectory $\mathbf{s}_{\text{traj}}$, goal trajectory $\mathbf{g}_{\text{traj}}$, and diffusion step $k$ as input, and predicts the added noise as $\epsilon_\theta$. The predicted noise $\epsilon_\theta$ is then regressed to match the true noise $\epsilon_k$ with mean square error loss. In this way, the denoising model is learned to generate sequences of low-level actions conditioned on robot states and goals from the dataset. Details of the model architecture and training objective are introduced in Section 4.

**Deployment**  In the last stage, we zero-shot transfer the trained DiffuseLoco policy to the robot hardware. During deployment, the DiffuseLoco policy takes a sequence of pure noise sampled from a Gaussian distribution and denoises it conditioned on states $\mathbf{s}_{\text{traj}}$ and the goal $\mathbf{g}_{\text{traj}}$. The denoising process is repeated for $K$ iterations to generate a sequence of actions, but only the *immediate* action $\mathbf{a}_t$ is used as the robot's joint-level PD targets. After executing this action, the policy takes a new sequence of states from the robot and updates the immediate action from the newly generated sequence. This aligns with the Receding Horizon Control (RHC) framework, instead of interpolating the action sequence at high frequency in prior works [50, 60]. RHC allows DiffuseLoco to replan quickly with fast-changing robot states to ensure up-to-date actions while considering future steps. However, due to the large number of parameters and denoising steps, we must accelerate inference to meet the RHC control frequency. The acceleration techniques are detailed in Appendix G, which enable running the DiffuseLoco policy on an edge-compute device mounted on the robot.

## 4 Diffusion Model for Real-Time Control

In this section, we introduce the diffusion model backbone, shown in Figure 2b, with a special focus on design choices for real-time control and inference acceleration.

**DDPM for Control**  To model multi-modal behaviors from diverse datasets, we leverage Denoising Diffusion Probabilistic Models (DDPM) [61] with a transformer backbone. DDPM is a class of generative models where the generative process is modeled as a denoising procedure, often referred to as Stochastic Langevin Dynamics, expressed in the following equation,

$$\mathbf{x}^{k-1} = \alpha \left( \mathbf{x}^k - \gamma \epsilon_\theta(\mathbf{x}^k, k) + \mathcal{N}(0, \sigma^2 I) \right) \tag{1}$$

where $\mathcal{N}(0, \sigma^2 I)$ denotes the sampled noise, $\alpha$, $\gamma$, and $\sigma$ are its hyperparameters. For clarity, we now use subscripts $_{t-a:t-b}$ to indicate trajectories from timestep $t - a$ to $t - b$. First, an initial noisy action sequence, $\mathbf{a}_{t:t+n}^K$, is sampled from Gaussian noise, and the DDPM conditioned on states $\mathbf{s}_{t-h:t}$, goals $\mathbf{g}_{t-h:t}$, and previous actions $\mathbf{a}_{t-h-1:t-1}$ undergoes $K$ iterations of denoising steps.

Unlike previous works applying DDPM in manipulation [49], the inclusion of previous actions, *i.e.*, I/O history, is crucial to better perform system identification and state estimations for dynamic legged control, as evaluated in [18]. Furthermore, instead of concatenating state and goal into a single embedding [49, 50], we leverage the transformer's attention mechanism to assign different attention weights to separately embedded rapidly-changing I/O history and relatively static goals, enabling the policy to adjust focus between adapting to dynamic environments and achieving goals.

The denoising process yields a sequence of intermediate actions characterized by progressively decreasing noise levels: $\mathbf{a}^K, \mathbf{a}^{K-1}, \ldots, \mathbf{a}^0$, until the desired noise-free output, $\mathbf{a}^0$, is attained. This process can be expressed as the following equation:

$$\mathbf{a}_{t:t+n}^{k-1} = \alpha(\mathbf{a}_{t:t+n}^k - \gamma\epsilon_\theta(\mathbf{a}_{t-h-1:t+n}^k, \mathbf{s}_{t-h:t}, \mathbf{g}_{t-h:t}, k) + \mathcal{N}(0, \sigma^2 I)) \tag{2}$$

where $\mathbf{a}_{t:t+n}^k$ represents the output at the $k^{\text{th}}$ iteration, and $\epsilon_\theta(\mathbf{a}_{t-h-1:t+n}^k, \mathbf{s}_{t-h:t}, k)$ represents the predicted noise from the denoising model $\epsilon_\theta$, which is parameterized by $\theta$, with respect to $\mathbf{a}_{t-h-1:t+n}^k$, $\mathbf{s}_{t-h-1:t}$, and iteration $k$.

During training, we opt to use the simplified training objective proposed by Ho et al. [61],

$$l = MSE\left(\epsilon_k, \epsilon_\theta(\mathbf{a}_{t-h-1:t+n} + \epsilon_k, \mathbf{s}_{t-h:t}, \mathbf{g}_{t-h:t}, k)\right). \tag{3}$$

where $\epsilon_k$ is the sampled noise at iteration $k$. Appendix B describes further architecture details.

**Delayed Input and Predicted Actions** To achieve real-time deployment, we predict current actions using delayed inputs, overcoming the long inference times of large models like transformers that exceed the 30Hz frequency requirement.

In DiffuseLoco, we use one-step delayed inputs—$\mathbf{s}_{t-h-1:t-1}$, $\mathbf{a}_{t-h-2:t-2}$, and $\mathbf{g}_{t-h-1:t-1}$—to predict current actions $\mathbf{a}_{t:t+n}$. Initiating inference before receiving the current state $\mathbf{s}_t$ allows parallel processing and ensures up-to-date actions. We adopt this design for DiffuseLoco because it learns sequences of action predictions, suitable for generating actions further in time, and handles higher input delays better than small-scale MLP policies, which typically manage delays less than one control step, for instance, 25% less than DiffuseLoco as noted in [18, Table IV].

**Sampling Techniques** To accelerate diffusion models during robotic deployment, prior work often uses samplers like the Denoising Diffusion Implicit Models (DDIM)[62], which employ a deterministic process to reduce sampling steps and speed up inference, albeit with some quality loss. However, in real-time legged control, DDIM's less accurate outputs increase step-by-step errors, leading to higher compounding errors and more out-of-distribution scenarios during deployment. Thus, we adopt DDPM for its enhanced robustness and performance in real-time control. The ablation can be found in Appendix D.3.

## 5 Results: Model Capacity

To scale up learning locomotion skills as discussed in Section 1, the critical questions are whether DiffuseLoco can (a) be trained with various sources of demonstrations and (b) incorporate a diverse set of skills present in the dataset. We answer these questions by presenting a state-of-the-art five-skill policy that combines four quadrupedal skills, and more importantly, a bipedal locomotion skill for quadrupedal robots, not yet demonstrated by prior RL frameworks.

**Learning from Diverse Data Sources** Figure 3 illustrates the five skills DiffuseLoco acquires for a quadrupedal robot: trotting and pacing (trained with AMP [32]), hopping and bouncing (trained with nominal CPG curves [27]), and agile bipedal walking (trained with symmetry augmentation [63]). After collecting demonstrations of these skills separately in simulation, we directly learn from this combined dataset and achieve robust zero-shot transfer to actual hardware. This capability surpasses previous offline learning methods that were largely confined to simulated environments with simplified dynamics, showcasing DiffuseLoco's robust real-world performance.

The ability to learn from diverse skill sources is crucial for scaling locomotion learning. For instance, bipedal walking requires specific early termination conditions and reward landscapes, unlike quadrupedal skills. Also, while basic symmetry augmentation yields effective motions, prior methods like AMP [12, 36] require complex trajectory optimization as reference, posing significant challenges. Despite different requirements for observation, action spaces, and auxiliary signals across RL methods, DiffuseLoco is able to accomplish all skills with only basic proprioceptive inputs.

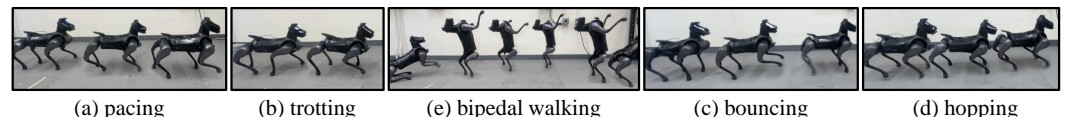

| (a) pacing | (b) trotting | (e) bipedal walking | (c) bouncing | (d) hopping |

Figure 3: Snapshots of five diverse agile locomotion skills with a single DiffuseLoco policy. This represents a leading effort in developing a single policy that can combine an agile bipedal walking skill with other quadrupedal skills and can be deployed on real-world robots.

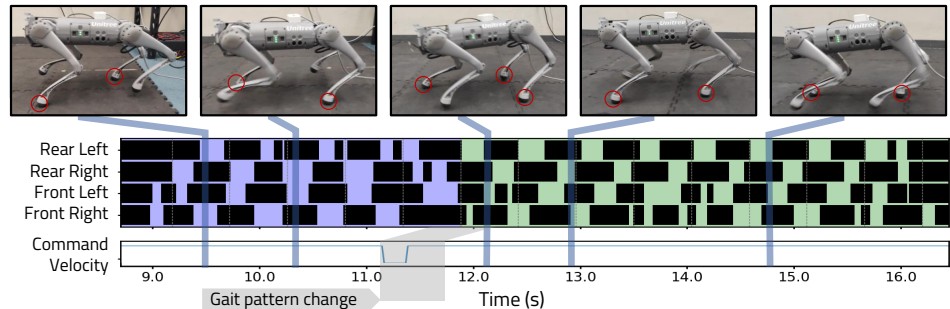

Figure 4: Demonstration of stable walking with different modalities. The red circle denotes the legs in contact with the ground. The robot initially walks using trotting skill, shown in purple background, then switches to pacing, shown in green, following a command change that involves a sudden stop and resume. We emphasize DiffuseLoco's ability to maintain different modalities under the same command, switching only when necessary.

**Skill Transitioning** We emphasize DiffuseLoco's capacity to transition freely between skills *without* transition data in the dataset, such as from hopping to bipedal walking and then to pacing, as shown in Figure 1. This sequence highlights DiffuseLoco's robustness against variations in starting state and the stability required to execute these skills successfully. Additional skill transitions are available in Appendix A. We also analyze quantitatively on transition smoothness in Appendix C.

In addition to transitions under *different* goals (commands), DiffuseLoco also demonstrates both trotting and pacing under the *same* goal. As shown in Figure 4, the policy begins with trotting and only switches to pacing when a sudden braking event significantly alters the contact sequence. This highlights DiffuseLoco's effectiveness in learning and adhering to different modes from the offline dataset, committing to a single mode within each rollout unless prompted by external disturbances.

**Extension to Bipedal Robots** In addition to quadrupedal robots, we also demonstrate the effectiveness of our method on high-dimensional, highly non-linear human-sized bipedal robot in the MuJoCo simulation [64]. First, we collect demonstrations evenly from two separately trained single-skill RL policies on walking and running, adapted from [18]. After training directly on this aggregated dataset, our method successfully learns both skills within a single policy. Furthermore, as shown in Figure 1, our policy can transition from walking to running smoothly without specific transition data in the dataset, in addition to maintaining each skill's stability before and after transitions. This demonstrates one of the initial working combinations of these skills on bipedal robots.

**Robustness** To demonstrate DiffuseLoco's robustness, we show both quadrupedal and bipedal locomotion over different ground conditions, including padded floor, bare floor, turf, and over small terrain variations, in Figure 5. Notably, bipedal walking over a half-padded floor shows high robustness to differences in ground height, friction, and restitution forces on each of the standing legs. Compared to DAgger, DiffuseLoco captures the randomized dynamics via offline data only.

In conclusion, we demonstrate that DiffuseLoco can learn a diverse set of skills from various offline sources and specialized RL frameworks. Importantly, DiffuseLoco shows better scalability in learning diverse and agile locomotion skills that existing RL frameworks have not yet illustrated. It also demonstrates robustness, skill transitions, and the ability to extend to more complex legged robots.

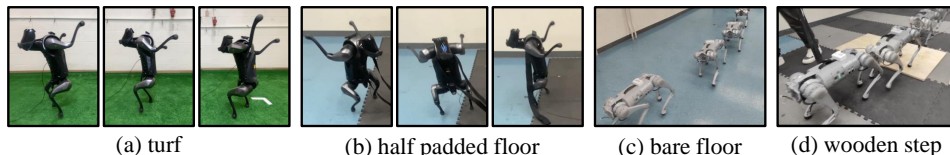

| | (a) turf | (b) half padded floor | (c) bare floor | (d) wooden step |

Figure 5: DiffuseLoco's robustness to grounds and terrains: bipedal walking on (a) turf, (b) half padded floor, where the ground heights, friction and restitution on the two standing legs are different; quadrupedal walking on (c) bare floor, (f) over a thick wooden board as a variation in terrain height.

| Goal (Task) | Metric | AMP | AMP w/ H | TF | TF w/ RHC | DiffuseLoco (Ours) |
|---|---|---|---|---|---|---|
| $0.3 m/s$ **Forward** | Stability (%) | **100** | **100** | 80 | **100** | **100** |
| | $E_v$ (%) | $90.44 \pm 1.87$ | $90.63 \pm 4.79$ | $75.75 \pm 6.07$ | $39.28 \pm 2.34$ | $\mathbf{33.22 \pm 12.48}$ |
| $0.5 m/s$ **Forward** | Stability (%) | **100** | **100** | **100** | **100** | **100** |
| | $E_v$ (%) | $50.44 \pm 1.97$ | $46.29 \pm 2.55$ | $54.35 \pm 2.66$ | $37.46 \pm 5.31$ | $\mathbf{12.91 \pm 6.84}$ |
| $0.7 m/s$ **Forward** | Stability (%) | 0 | 20 | 0 | 40 | **100** |
| | $E_v$ (%) | fail 5/5 | $54.96 \pm 0.00$ | fail 5/5 | $39.36 \pm 5.02$ | $\mathbf{24.80 \pm 8.91}$ |
| **Turn Left** | Stability (%) | 20 | **100** | 0. | **100** | **100** |
| | $E_v$ (%) | $20.96 \pm 0.00$ | $33.39 \pm 6.96$ | fail 5/5 | $13.41 \pm 5.02$ | $\mathbf{12.79 \pm 5.64}$ |
| **Turn Right** | Stability (%) | **100** | **100** | **100** | 80 | **100** |
| | $E_v$ (%) | $18.61 \pm 2.40$ | $33.39 \pm 6.96$ | $25.86 \pm 1.47$ | $8.69 \pm 5.04$ | $\mathbf{2.22 \pm 1.03}$ |

Table 1: Benchmark for baselines and DiffuseLoco in the real world. Stability (higher is better) measures the percentage of trials where the robot remains stable without falling. $E_v$ (lower is better) measures the percentage deviation from the desired velocity. Experiments are conducted with different commands (left), each repeated five times, and the mean and standard deviation are reported.

## 6 Results: Quantitative Analysis

In this section, we first compare DiffuseLoco against multi-skill RL (AMP) and non-diffusion BC baselines. Since there is no RL baseline with all five skills, we focus on pacing and trotting in this analysis. Next, we ablate on the key model and dataset characteristics, such as model size and skill size to study their impact on performance. Detailed task setups are provided in Appendix E.

**DiffuseLoco versus AMP (RL)**    We first compare DiffuseLoco with RL-based multi-skill control policy **AMP**. Table 1 shows that **DiffuseLoco** is the only method among the baselines that is able to reliably complete all trials without falling over. Specifically, the RL-trained **AMP** and **AMP with history inputs** (**AMP w/ H**) baselines struggle with low and high speed commands. For 0.3 m/s forward command, the actual velocity is more than 90% slower than the commanded velocity. For 0.7 m/s forward command, they achieve a stability metric of 0% and 20% respectively.

This shows the prevailing problem of mode-collapse for Generative Adversarial Networks (GAN), where the generator becomes overfitted to a limited range of outputs that are often similar or identical [65, 66, 67]. In AMP, the actor overfits the simulation environment, losing generalization to new environments (*e.g.* real world), but reverting to early stage behaviors where the discriminator is not yet converged. An example is in low-speed tasks where it oscillates in place with correct frequency but reduced amplitude, similar to its behaviors in early training stage.

In contrast, **DiffuseLoco** generalizes within the expert demonstrations and does not revert to infeasible actions. Moreover, **DiffuseLoco** with its diffusion model efficiently learns the multi-modality of the different skills for the same command, enabling it to perform valid locomotion skills without mode-collapsing. In general, **DiffuseLoco** achieve both better stability and velocity tracking performance compared with AMP-based baselines.

**DiffuseLoco versus Non-Diffusion Behavior Cloning**    We further compare **DiffuseLoco** with non-diffusion BC methods, such as transformer with one-step prediction (**TF**) or with a prediction horizon (**TF w/ RHC**). Shown in Table 1, **DiffuseLoco** outperforms **TF** and **TF w/ RHC** in both stability and robustness. Specifically, **TF** lacks robustness and fails the 0.7 m/s forward and left turn tasks completely, likely because single-step action prediction makes the policy less aware of future actions, leading to more jittering behavior.

With receding horizon control, **TF w/ RHC** overcomes most of the jittering problem and can complete most of the tasks. However, for more agile motion such as 0.7 m/s forward, the stability metric drops drastically to merely 40%, likely due to the reconstruction loss used in **TF w/ RHC** training tends to overfit the action trajectories in the dataset, resulting in less robust policy in the out-of-distribution scenarios (*e.g.* real world).

In comparison, our **DiffuseLoco** shows more stable and smooth motions measured by both stability metrics and magnitudes of the body's angular velocity. On average, **DiffuseLoco** achieves 10.40% less in magnitude for the body's oscillation over all trials. As a result, the smoother locomotion skill leads to on average 38.97% less tracking error compared to **TF w/ RHC**. Thus, we believe diffusion-style training is more suitable for locomotion tasks than previous Behavior Cloning methods.

| Ablate on | Num. of Predicted Future Actions | | | | Model Parameter Count | | | | Num. of Dataset Skills | |
|---|---|---|---|---|---|---|---|---|---|---|
| | 1 | 4 | 8 | 16 | 120K | 600K | 1.2M | 6.8M | 1 | 5 |
| **Live Time (s)** | $11.51 \pm 0.6$ | $14.21 \pm 0.4$ | **$15.34 \pm 0.3$** | $13.69 \pm 0.2$ | $0.94 \pm 0.0$ | $6.31 \pm 0.3$ | $13.51 \pm 0.1$ | **$14.21 \pm 0.4$** | $13.11 \pm 0.4$ | **$14.21 \pm 0.4$** |
| **% Timeouts** | $47.5 \pm 3.6$ | $63 \pm 1.4$ | **$69.5 \pm 2.1$** | $58.5 \pm 3.5$ | $0 \pm 0.0$ | $22 \pm 1.4$ | $56.5 \pm 0.7$ | **$63 \pm 1.4$** | $56.5 \pm 3.5$ | **$63 \pm 1.4$** |

Table 2: Ablation results on key characteristics. A medium length of action predictions yields the best performance. For model and skill sizes, DiffuseLoco scales with larger models and more skills.

**Ablations**  We perform ablations to study the key characteristics. We report the average live time and the percentage of episode timeouts (success) on bipedal walking across 100 randomized environments with 20-second maximum length. As shown in Table 2, predicting only the next action leads to jittery movements and worse performance, while predicting too far into the future (16 steps) also degrades performance due to higher variance in future actions. The optimal performance is achieved with moderate prediction steps. For model size, performance consistently improves with larger parameter counts, confirming that DiffuseLoco scales with model capacity, and the large expressive model is necessary to handle the challenging five skills. For number of skills, we compare the multi-skill policy with a single-skill policy (bipedal walking) trained on the same shard of bipedal data. We find that multi-skill training slightly improved performance, likely because it exposes the policy to a broader range of dynamics, enhancing generalization and robustness. Thus, DiffuseLoco scales with increasing number of skills. Further ablations on design choices are in Appendix D.

# 7   Limitation

Admittedly, DiffuseLoco's robustness is subpar compared to the state-of-the-art single-skill RL. For instance, the sim-to-real gap is larger on bipedal hardware due to more complex dynamics requiring more accurate data. Fortunately, DiffuseLoco can learn directly from offline real-world data without accessing expert policies or repeated data collection. Appendix D.5 shows the ablation of improving the robustness by absorbing data from various dynamics. By expanding our dataset with more diverse, potentially real-world data, we anticipate its robustness will improve progressively.

# 8   Discussion and Future Work

We have present DiffuseLoco, a scalable framework of state-of-the-art performance in learning diverse legged locomotion skills from multi-modal *offline* datasets with robust transfer to hardware and real-time inference. Leveraging diffusion models, DiffuseLoco learns bipedal and quadrupedal skills within one policy and transitions freely among them without extra transition data. It absorbs demonstrations from various RL algorithms with different observation and action spaces and learns different modalities under identical commands. Furthermore, it offers a practical inspiration for scalable real-world data collection and learning. Additionally, in future work, DiffuseLoco could adapt to datasets with different robot morphologies and integrate vision and language instructions, enhancing its versatility and applicability. The five diverse skills presented in this work demonstrate DiffuseLoco's scalability towards a generalist policy for legged locomotion control.

## 9 Acknowledgement

This work was supported in part by NSF 2303735 for POSE, in part by NSF 2238346 for CAREER, in part by The AI Institute and in part by InnoHK of the Government of the Hong Kong Special Administrative Region via the Hong Kong Centre for Logistics Robotics.

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

# A More Results on Skill Transitioning

In this section we present more results on a series of video clips to showcase skill transitioning of DiffuseLoco.

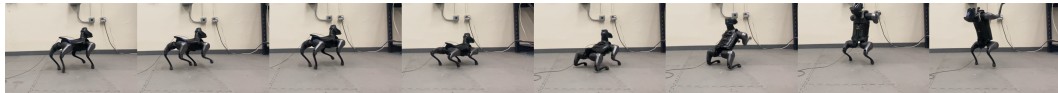

Figure 6: Skill Transitioning: Pace to Stand

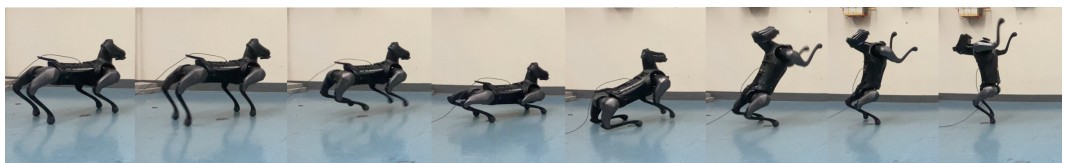

Figure 7: Skill Transitioning: Hop to Stand on Bare Floor

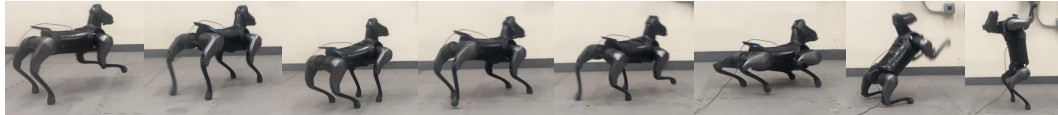

Figure 8: Skill Transitioning: Bounce to Stand with Emergent Intermediate Pacing Skill

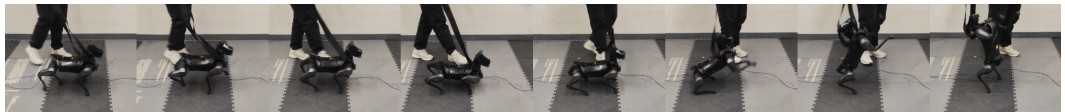

Figure 9: Skill Transitioning: Trot to Stand

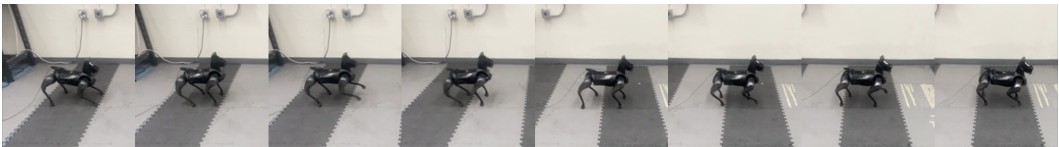

Figure 10: Skill Transitioning: Bounce to Pace

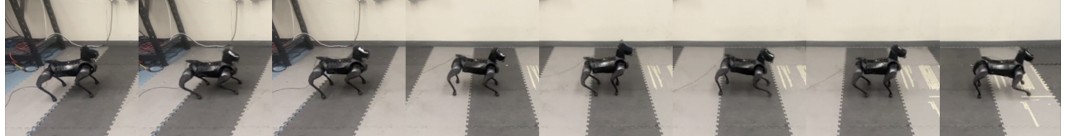

Figure 11: Skill Transitioning: Hop to Bounce

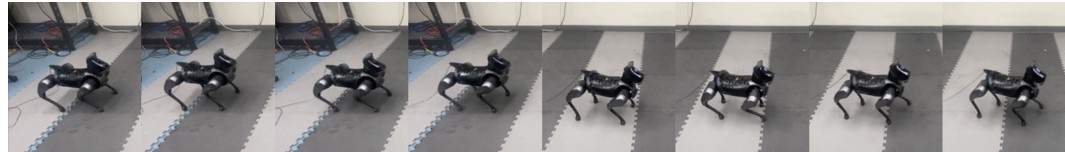

Figure 12: Skill Transitioning: Hop to Pace

# B Model Architecture and Hyperparameters

Here, we explain the details of the receding horizon control framework and diffusion model's architecture, dataflow, and transformer backboone.

## B.1 Receding Horizon Control

Learning sequences of actions instead of single-step action in training helps in improving temporal consistency of the policy. However, in dynamic systems such as legged robots, the error accumulates significantly after a short horizon of planned steps, and the predicted actions further ahead may

no longer be useful for control. Therefore, we adopt the Receding Horizon Control (RHC) manner, where DiffuseLoco policy generates $n$ steps of actions, but only executes the *very first step* of actions. This is in contrast to previous work which infers a sequence of actions at a lower frequency and uses interpolation to get a high-frequency action [60, 50]. Such a setup allows us to replan rapidly with fast-changing states of the robot while keeping future steps in account. As we evaluated in Appendix D.2, using RHC is critical in improving smoothness and consistency of a legged locomotion control policy.

## B.2 Architecture

The DiffuseLoco policy leverages an encoder-decoder transformer DDPM. First, the past robot's past I/O trajectory $(\mathbf{s}_{t-h-1:t-1}, \mathbf{a}_{t-h-2:t-2})$ and given goal sequence $\mathbf{g}_{t-h-1:t-1}$ are transformed into separate I/O embedding and goal embedding by two 2-layer MLP encoders, respectively. Then, we sample noise $\epsilon(k)$ for diffusion time step $k$ with the DDPM scheduler and add to the ground truth action $\mathbf{a}$ from the offline dataset to produce a noisy action $\mathbf{a}^k_{t:t+n} = \mathbf{a}_{t:t+n} + \epsilon_k$. The noisy action $\mathbf{a}^k_{t:t+n}$ is then passed through an MLP layer into action embedding. The noisy action tokens are then passed through 6 Transformer decoder layers, each of which is composed of an 8-head cross-attention layer. Each layer computes the attention weights for the noisy action tokens querying all the state embedding, goal embedding, and the timestep embedding reflecting the current diffusion timestep $k$. We apply causal attention masks to each of the state embeddings and goal embeddings separately. The predicted noise $\epsilon_\theta(\mathbf{a}_{t-h-2:t+n}, \mathbf{s}_{t-h-1:t-1}, \mathbf{g}_{t-h-1:t-1}, k)$ is then computed by each corresponding output token of the decoder stack. We then supervise the output to predict the added noise with Eqn. 3 to find optimal parameters $\theta$ of the denoising model $\epsilon_\theta$.

## B.3 Delayed Inputs and Predicted Actions

Using delayed inputs is uncommon in RL-based locomotion control but is necessary here due to the longer inference time of diffusion models compared to MLPs. Unlike prior works (mostly based on RL) which commonly consider the zero-order-hold *communication delay* and ignore the *inference delay*, DiffuseLoco's inference takes 10-20ms on different GPUs (50-100% of a control cycle), making the latter much more significant. To demonstrate the impact of this delay, we run an ablation without delayed inputs and test with inference delay. With 10ms inference (on the most recent RTX 4090), the robot continued walking with minor staggering, but at 15ms, it cannot stand up anymore.

## B.4 Hyperparameters

The hyperparameters are summarized in Table 3,

| | Five-Skill (Sec. 5) | Walk (Sec. 6) | Cassie (Sec. 5) |
|---|---|---|---|
| History Length | 8 | 8 | 16 |
| Predict Length | 4 | 4 | 4 |
| Token Dim | 256 | 128 | 256 |
| Attn Drop-out | 0.3 | 0.3 | 0.3 |
| Learning Rate | 1e-4 | 1e-4 | 1e-4 |
| Weight Decay | 1e-3 | 1e-3 | 1e-3 |
| Epochs | 100 | 100 | 100 |

Table 3: Hyperparameters for Diffuse-Loco in the Experiments

| Phase | Mean | STD | Max | Min |
|---|---|---|---|---|
| Hopping | 0.52 | 0.49 | 2.31 | 0.02 |
| Transition | 0.53 | 0.42 | 1.48 | 0.005 |
| Pacing | 0.75 | 0.44 | 1.80 | 0.03 |

Table 4: Base angular velocity statistics during skill transitioning from hopping to pacing.

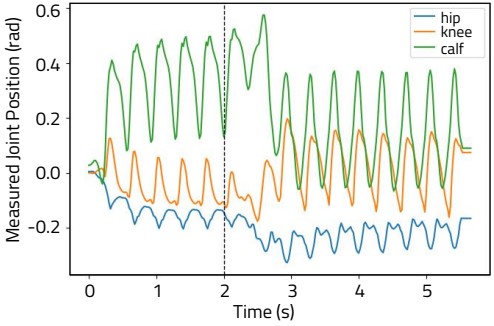

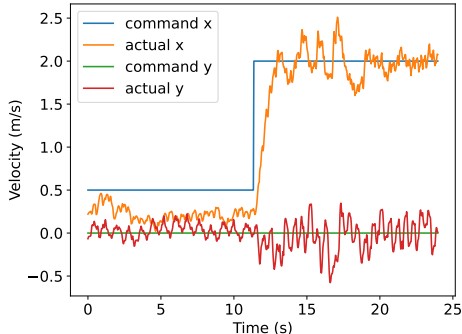

Figure 13: Joint positions during transition from hopping to pacing. The vertical dashed line marks the time of receiving the pacing command.

Figure 14: Bipedal robot skill transitioning. Velocity tracking with transition from walking to running on the bipedal robot Cassie in Mujoco simulation.

## C  Quantitative Analysis on Transition Smoothness and Bipedal Robot

### C.1  Quantitative Analysis on Transition Smoothness

As shown in Figure 4, the mean and standard deviation (STD) of base angular velocity during skill transitions are similar to steady single-skill execution, indicating comparable body oscillation. The maximum value during transitions does not exceed that of single-skill execution, suggesting stable body dynamics. These statistics come from a single trial, with STD reflecting angular velocity fluctuations over time. A higher STD indicates more fluctuations, which is undesirable. As some fluctuation is expected during locomotion, an STD around 0.4 is normal. Transitions, like from hopping to trotting, may involve acceleration and deceleration, so the mean value may not simply be an interpolation. We define smooth transitions as having minimal abrupt oscillations and consistent angular velocity, though interpretations may vary.

In Figure 13 , we present the joint position trajectories during the skill transition from hopping to pacing. The plot illustrates that after receiving the new command (indicated by the vertical dashed line), the policy does not switch to the pacing mode immediately. Instead, it completes the current gait cycle before transitioning. During the subsequent cycle, the system interpolates from the current joint position to a suitable starting position for the new skill. Following this interpolation, the system initiates the pacing gait and gradually adjusts the joint movement magnitudes from the interpolated state towards steady-state execution. We note that naively switching between different skill policies would lead to abrupt changes, as the system would immediately begin executing the new skill without proper interpolation.

### C.2  Quantitative Result on Bipedal Robot

In Figure 14, we show a plot of the command velocity v.s. actual velocity on bipedal robot Cassie in Mujoco simulation involving a transition from walking to running. Although some steady-state error exists in low-speed walking, we see that the robot is able to achieve and maintain a steady running speed of 2.0m/s as commanded.

## D  Ablation Study on Design Choices

Here, we further evaluate the design choices used to build DiffuseLoco policy in simulation and the real world by extensive ablation studies. We use the same experiment setups as in Section 6, detailed in Appendix E.

| Goal (Task) | Metric | DL w/o RHC | DL w/o Rand | DDIM-100/10 | DDIM-10/5 | U-Net | DiffuseLoco (Ours) |
|---|---|---|---|---|---|---|---|
| $0.3m/s$ **Forward** | Stability (%) | **100** | **100** | **100** | **100** | **100** | **100** |
| | $E_v$ (%) | $75.09 \pm 18.98$ | $50.45 \pm 2.70$ | $56.89 \pm 2.43$ | $47.09 \pm 2.40$ | $81.31 \pm 1.90$ | $\mathbf{33.22 \pm 12.48}$ |
| $0.5m/s$ **Forward** | Stability (%) | **100** | 80 | 80 | **100** | **100** | **100** |
| | $E_v$ (%) | $64.49 \pm 1.87$ | $41.07 \pm 6.12$ | $41.00 \pm 3.18$ | $37.92 \pm 1.59$ | $74.52 \pm 2.83$ | $\mathbf{12.91 \pm 6.84}$ |
| $0.7m/s$ **Forward** | Stability (%) | 0 | 40 | 80 | 80 | **100** | **100** |
| | $E_v$ (%) | fail 5/5 | $44.30 \pm 4.21$ | $47.71 \pm 6.63$ | $42.58 \pm 2.08$ | $71.71 \pm 2.93$ | $\mathbf{24.80 \pm 8.91}$ |
| **Turn Left** | Stability (%) | **100** | **100** | **100** | **100** | 20 | **100** |
| | $E_v$ (%) | $20.96 \pm 18.22$ | $\mathbf{10.17 \pm 5.86}$ | $22.22 \pm 4.29$ | $13.27 \pm 2.63$ | $18.93 \pm 23.28$ | $12.79 \pm 5.64$ |
| **Turn Right** | Stability (%) | **100** | **100** | **100** | **100** | **100** | **100** |
| | $E_v$ (%) | $18.61 \pm 2.40$ | $8.18 \pm 3.94$ | $6.47 \pm 2.49$ | $7.42 \pm 2.90$ | $89.63 \pm 3.36$ | $\mathbf{2.22 \pm 1.03}$ |

Table 5: Performance Ablation Study across different ablations and DiffuseLoco policy in real-world experiments. Stability (the higher the better) measures the number of trials in which the robot stays stable and does not fall over. $E_v$ (the lower the better) measures the deviation from the desired velocity in percentage. The experiments are conducted with different command settings (Left). Each command is repeated non-stop for five trials, and we report the average and standard deviation of the metrics across five trials.

## D.1 Ablation Components

To validate our design choices, we ablate **DiffuseLoco** with the following critical components and compare them to our real-world benchmark.

- Without Receding Horizon Control (**DL w/o RHC**): Replace RHC with one-step prediction in an autoregressive manner and keep the diffusion model.

- Without Domain Randomization (**DL w/o Rand**): Trained on a dataset generated without domain randomization, except for the ground friction coefficient.

- DDIM Inference: We develop two DDIM baselines to investigate how training and inference steps affect performance in locomotion control.

   - 100 Training + 10 Inference (**DDIM-100/10**)
   - 10 Training + 5 Inference (**DDIM-10/5**)

   Compared with our **DiffuseLoco**, **DDIM-100/10** has the same inference steps, and **DDIM-10/5** has the same training steps.

- U-Net as the backbone (**U-Net**): Replace the Transformer with a U-Net as the backbone, adjusted to the same parameter count.

## D.2 Single-step output versus RHC

To isolate the effects of RHC, we test a variant of DiffuseLoco without RHC (**DL w/o RHC**), finding that it struggles with faster speed goals and exhibits significant jittering behaviors, as detailed in Table 5. This suggests that single-step token-prediction models like GPT are less suitable for legged locomotion control than diffusion models, which predict sequences of future actions.

## D.3 Sampling Techniques

As discussed earlier, popular diffusion-based frameworks like DDIM often reduce sample iterations for inference acceleration, trading off output quality for speed, often with ten times fewer iterations [62]. While this approach suits tasks like image generation, which tolerate some variance, it underperforms in quadrupedal locomotion control. As shown in Table 5, both **DDIM-100/10** and **DDIM-10/5** exhibit worse stability and higher velocity tracking errors. Noticeably, the 100 training steps and 10 inference steps variant demonstrates limping behavior and fails two trials. Tracking errors for both variants increase by 50.69% and 42.04%, respectively, compared to **DiffuseLoco**.

Thus, we believe that noisier control signals from the DDIM pipeline likely disrupt the control of inherently unstable floating-based dynamic systems, like legged robots. An interesting future work direction could be on control-specific sampling techniques to accelerate diffusion models without compromising stability and performance.

## D.4    Model Architecture Effects

In addition, we compare against another commonly used architecture in diffusion models, a CNN-based U-Net as the backbone of DiffuseLoco. Qualitatively, the **U-Net** policy is shaky and inconsistent, and quantitatively, it has one of the highest errors in velocity tracking, with worsened stability due to its shaky actions. We reckon that this is because CNNs are not the best fit for temporal data and also lack separate attention weights for goal conditioning. This is consistent with prior work [49] that finds **U-Net** underperforms Transformer, especially in high action-rate dynamical systems.

## D.5    Dataset Effects

Lastly, we explore how dataset characteristics influence the robustness and performance of **Diffuse-Loco** in real-world scenarios. Consistent with previous findings that diversity in training data, such as noise insertion, mitigates compounding error [68], we demonstrate that increasing the variety of dynamics parameters in simulation environments where we collect data also enhances robustness. As in Table 5, training DiffuseLoco on a dataset with dynamics randomization leads to a 44.26% increase in both robustness and stability compared to **DL w/o Rand** baseline. Specifically, in the challenging 0.7 m/s forward task, **DL w/o Rand** falls in 3 out of 5 trials. This ablation study points to the potential of altering the dataset, by adding either more diversity and potentially real-world data or more fault-recovery behaviors, to further enhance the robustness of DiffuseLoco.

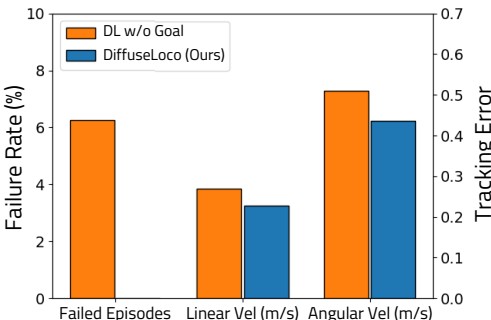

Figure 15: Comparison of failure rates and tracking errors between **DL w/o Goal** and **DiffuseLoco** (ours) in simulation. The left y-axis is the metric for Failed Episodes. The right y-axis indicates the tracking error for linear velocity and angular velocity.

## D.6    Use of Goal-conditioning

Here, we evaluate the impact of goal-conditioning, hypothesizing enhancements tracking performance and stability. In the **DL w/o Goal** baseline, we do not add the goal-conditioning encoder. Instead, the goal is concatenated with the robot's I/O. Considering the noisy base velocity estimations on real robots, we utilize simulation environments with extensive dynamics randomization to perform a large number of trials and get more systematic results. As shown in Figure 15, Diffuse-Loco achieves a 15.4% reduction in linear velocity tracking error and a 14.5% reduction in angular velocity error compared to the **DL w/o Goal** baseline. Moreover, over 64 trials with identical commands, **DL w/o Goal** falls over four times, or 6.25% of all trials, whereas DiffuseLoco experiences no failures. This pattern persists in real-world testing, where **DL w/o Goal** fails one trial in a 0.7m/s forward test.

These results underscore the importance of goal-conditioning with distinct attention weights for dynamic system control, revealing that the robot's I/O history and goals, governed by physics and arbitrary objectives respectively, should not be merged into one embedding space.

### D.7  Finetuning DiffuseLoco

We find that a pre-trained policy indeed provides a solid starting point to finetune other skills with efficient samples. For example, after training on 500 episodes of only quadrupedal trotting skill, we finetuned this policy on only 10 episodes, 10,000 steps (200-second long) in total, of hopping data and found that 10 episodes were enough for the policy to learn this new skill. In terms of using real-world data to finetune with the flat-terrain policy, we expect we can collect a small number of trajectories collected in the real world to finetune the policy. We mark finetuning with real-world data as an interesting and feasible future work.

## E  Experiment Setup (for Section 6 and Appendix D)

### E.1  Task and Baselines

This analysis includes walking for four meters under five goals (commands) with different velocities. The goals are the following: move forward at three different speeds: $0.3 \ \mathrm{m/s}$, $0.5 \ \mathrm{m/s}$, and $0.7 \ \mathrm{m/s}$, and make a left turn and a right turn at $0.3 \ \mathrm{rad/s}$. We record the actual linear velocities via a Kalman filter state estimation [69], and the number of trials where the robot does not fall over through out the trial as the stability metrics. We repeat each experiment five times consecutively and report the mean and standard deviation across five runs.

Skill information are often unscalable or unavailable during training and deployment. For a boarder range of applicability, we limit the scope of comparisons within not-skill-conditioned multi-skill RL and non-diffusion BC baselines. Specifically, the RL baselines include,

- Adversarial Motion Priors (**AMP**) [32]: An MLP policy trained using AMP with RL (PPO) and style reward of both pacing and trotting reference motions. We *directly* use the *open-sourced* checkpoint from [32]. We note that although several skill-conditioned RL policies [34, 35, 33] have been introduced since [32], yielding better sim-to-real results, progress in unconditioned multi-skill policies has been limited.

- AMP with history steps (**AMP w/ H**): To align with **DiffuseLoco**, we train an AMP policy with 8 steps of state and action history with the same setup as [32] and a similar evaluation return in simulation.

Furthermore, we compare **DiffuseLoco** with non-diffusion BC policies, which can be categorized into autoregressive token prediction [43, 70] and action sequence prediction as used in [71]. We adopt baselines for each category.

- Transformer with Autoregressive Token Prediction (**TF**): A Generative Pretrained Transformer (GPT) [72] policy similar to a decision transformer [43] without reward conditioning. This only generates one timestep action.

- Transformer with Receding Horizon Control (**TF w/ RHC**): A transformer policy with the same future step action predictions. The model's architecture is identical to our **DiffuseLoco** model, but it directly predicts future action sequences and the loss is replaced by the reconstruction loss $l = MSE(\pi_\theta(\mathbf{s}_t, \mathbf{g}_t), \mathbf{a}_t)$.

These baselines have the same parameter count of 6.8M and are trained with the same learning rate scheme and number of epochs as **DiffuseLoco**.

**Remark 1** *Typically, previous work uses DAgger style algorithms [73] to better cope with distribution shift, but these methods require access to the expert policy in training and online learning*

*environments. As a more scalable and versatile framework, we limit our focus to learning entirely from offline datasets.*

## F   Details in Offline Locomotion Dataset

After the architecture details, we introduce the details in creating the offline locomotion dataset used in this work. We will explain the state, goal, and action spaces, followed by a brief introduction to the source policies and dynamics randomization used to diversify the data. We collect a total of 4 million data of state-action-goal pairs in the offline dateset for the quadrupedal robot tasks, and 10 million transitions for the bipedal robot tasks.

### F.1   State Space

The state space is the robot's proprioceptive feedback. In the quadrupedal locomotion control case, this consists of the measured motor positions $\mathbf{q}_m$, measured motor velocities $\dot{\mathbf{q}}_m$, base orientation $\mathbf{q}_{\psi,\theta,\phi}$, and base angular velocities $\dot{\mathbf{q}}_{\psi,\theta,\phi}$. Note that we exclude quantities from the estimation of base velocity ($\dot{\mathbf{q}}_{x,y}$) to prevent additional estimation errors.

### F.2   Goal Space

The goal of the locomotion task is the commands given to the policy. For quadrupedal robots, the command includes desired sagittal velocity $q_x^d$ in the range of 0 m/s to 1 m/s, desired base height from 0.2 m to 0.6 m, and desired turning velocity $q_\psi^d$ from -1 rad/s to 1 rad/s.

### F.3   Action Space

The action space is the robot's joint-level commands. In this work, we use the desired motor position $\mathbf{q}_m^d$ as the action. This is then used by joint-level PD controllers to compute motor torques $\tau$ at a higher frequency.

### F.4   Source Policy

We obtain source policies using three different RL methods. We leverage proximal policy optimization (PPO) [74] to optimize each of the source policies, and we train the policies in simulation (Isaac Gym [14]). We evenly distribute the data generated from each of the source skill-specific policies.

#### F.4.1   Adversarial Motion Prior (AMP)

For skills trained with AMP, we provide a reference motion retargeted from motion capture data of a dog [12], and incorporate an GAN-style discriminator to encourage the robot to imitate the reference motion without extended reward engineering. Then, the reward of this method is formulated as motion imitating term (provided by the discriminator [75]) and task term (*e.g.*, tracking error, etc). For fairness, we would like to highlight that the AMP baseline used in quantitative comparison in Section 6 and the data-generating AMP policies achieve *the same level* of performance in simulation.

#### F.4.2   Central Pattern Generator Guidance (CPG)

For skills trained with CPG-guidance, we follow the formulation in [76] and provide nominal reference motions of strictly periodic motions generated by a Central Pattern Generator with phase signals. Specifically, for hopping skill, the phase selections for all legs are 0, and for bouncing skill, the phase selections are 0 for the front legs, and $\pi$ for the hinder legs. The reward is composed of task term (*e.g.*, velocity tracking error, etc), motion tracking term (*e.g.*, reference motion tracking error, etc), and smoothing terms (*e.g.*, action rate, etc).

### F.4.3 Symmetry Augmented RL

We train the bipedal locomotion skill for quadrupedal robots following a symmetry-augmented RL policy [63] to achieve a symmetric gait pattern that is crucial for sim-to-real transfer. Specifically, the data collection process is augmented by the addition of symmetric states and actions. The reward includes task term (*e.g.*, velocity tracking error, etc), gait pattern term (*e.g.*, feet clearance height, etc), and smoothing terms (*e.g.*, action rate, etc).

### F.4.4 Other Methods

Although the source policies used to collect the dataset in this work are all RL-based policies, our framework is general and can include the data generated from model-based optimal controllers (such as from [77]) and others. The requirement is to align the state and action spaces among different source policies, and the frequency of the policy should be kept the same.

### F.5 Dynamics Randomization

In order to diversify the training dataset for DiffuseLoco, we also include the same amount of dynamics randomization [78] during the training of the source policies and the data generation using these policies. Specifically, in each episode in simulation, the dynamics parameters are randomized. These include the motor's PD gains, the mass of the robot's base (up to the weight of the onboard compute), ground frictions, and random changes in base velocity. The randomization ranges are adapted from the source policy's original methods.

## G  Real-Time Inference Acceleration

Although the diffusion model targets real-time use, it cannot meet the real-time targets without further tuning. Compared to previous works using Transformer for locomotion control with 2M parameters [79], our model is 3 times larger in parameter count (6.8M parameters) and 10 forward propagations are needed in each inference. Thus, an additional effort is needed to accelerate the diffusion on the edge computing device on the robot, such as the setup shown in Figure 17. In this section, we explore several methods to accelerate the inference process of the diffusion model to enable it to run real-time onboard.

### G.1  Acceleration Framework

Our DiffuseLoco policy has a parameter count of 6.8 million parameters, which exceeds most modern mobile processors' cache capacity. Furthermore, hardware on a typical consumer-grade central processing unit (CPU) is not optimized for the operators used in transformer networks. The graphics processing unit (GPU) is more suitable for computing the high-dimension matrix and vector operations. To ensure the portability of the setup, we use an accessible NVIDIA Mobile GPU as the deployment platform. For real-time deployment, an acceleration pipeline is built in the DiffuseLoco framework to convert and optimize our model towards the target compute platforms. The operators of the model are first extracted with ONNX [80]. Then, TensorRT is used to refine the execution graph and compile the resulting execution pipeline onto the target GPU. Through domain-specific architecture optimizations, the operations and memory access patterns are optimized to utilize the full capability of the GPU. With this approach, the speed for each denoising iteration is increased by about 7X compared to the native implementation in PyTorch, and the maximum inference (with 10 denoising iterations) frequency is increased from 17.0 Hz to 116.5 Hz. To showcase the effect of this acceleration approach, we conducted a benchmark on the inference frequency of the policy running on multiple hardware platforms we have access to, shown in Figure 16.

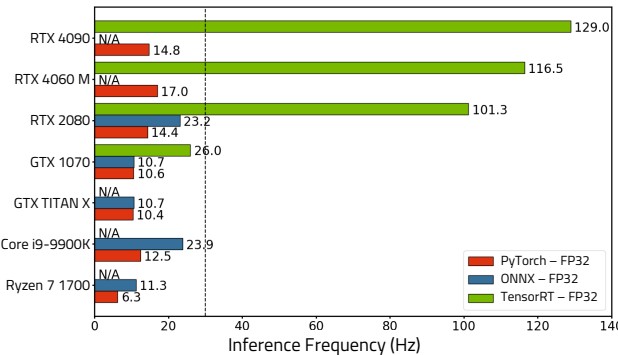

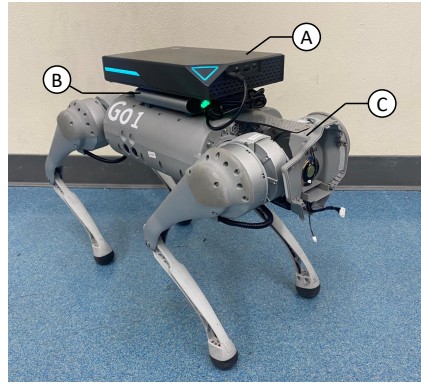

Figure 16: Benchmark of running our DiffuseLoco policy (6.8M parameters) on different hardware platforms. The dashed line marks the 30 Hz minimum frequency required for real-time robot control. TensorRT optimization achieves approximately 7x speedup compared to the naive PyTorch implementation. N/A entries reflect that the hardware is incompatible with the inference workflow.

Figure 17: Onboard compute experiment setup. A: Mini computer with Intel Core i7-13700H and NVIDIA GeForce RTX 4060 Mobile. B: Battery bank. C: Go1 quadrupedal robot. This setup is within the robot's load capacity, enabling real-time deployment of our diffusion model using the acceleration framework.

## G.2 Edge Compute for DiffuseLoco Policy on Robots

With the help of the acceleration framework, the compute platform can be deployed onboard a Go1 quadruped robot. A mini-computer equipped with Intel Core i7-13700H and NVIDIA GeForce RTX 4060 Mobile is attached to the top of the robot, as showcased in Figure 17. This computer runs DiffuseLoco policy and is powered by a dedicated battery bank, separated from the robot's internal battery. This arrangement is capable of running the policy for up to 90 minutes. The mini-computer connects to the robot via an Ethernet cable to send action for the joint-level PD controls on the robot's computer. We note that all the experiments we present are completed on the same edge computing device.

