# OpenReview forum: "DiffuseLoco: Real-Time Legged Locomotion Control with Diffusion from Offline Datasets"
_robot-learning.org/CoRL/2024/Conference — CoRL 2024_

### Official Review · Reviewer_TkLp · 2024-07-18

**Originality:** 3
**Technical Quality:** 3
**Clarity Of Presentation:** 3
**Potential Impact:** 3
**Recommendation:** 3
**Confidence:** 4

**Review:**

The authors follow the lines of prior work using diffusion models for imitation learning of manipulation skills. However, instead of using real-world datasets (e.g., from teleoperation), they use different sources from simulation, like training single-skill AMP policies. The authors show in various tasks and robots that their approach works well in simulation and on real hardware.

### Strengths:
- First to implement diffusion imitation policies for locomotion
- The paper is clearly written.
- strong empirical results on real hardware.

### Weaknesses:
- Some parts of the experiments remain unclear (see questions?)
- There does not seem to be any novel insight compared to diffusion for manipulation (Only tricks like delays typically used for locomotion policies)
- hence, no algorithmic contribution.

Since this paper does not provide an interesting new insight into the diffusion model for locomotion but rather a straightforward extension of manipulation work, I believe that the most significant contribution of this paper to the research community is the code base. Since the authors, are promising to provide the latter upon acceptance, I vote for acceptance.

**Quality Of The Limitations Section:**

3

**Questions For Rebuttal:**

### Questions:
- line 175: “Initiating inference before receiving the current state st allows parallel 175 processing and ensures up-to-date actions.” What exactly is processed in parallel?
- Line 184: “. Thus, we adopt DDPM for its en185 hanced robustness and performance in real-time control. “ What is adapted concretely? I didn’t find out even after the appendix? The appendix sounded more like a discussion?
- Line 189: “We answer these questions by presenting a state-of-the-art five190 skill policy that combines four quadrupedal skills, and more importantly, a bipedal locomotion skill 191 for quadrupedal robots, not yet demonstrated by prior RL frameworks.” In the related work are some works that did this already? → “[…] and walking on two feet with quadrupedal robots [20, 21, 22], requiring high agility and robustness. ”
- Section DiffuseLoco versus AMP (RL):
    - what exactly is a “skill”? Pacing and trotting ? Or each of them in a different direction and with a different velocity?
    - This is important to know, because it looks like your approach is surpassing AMP, which has been used to generate the dataset. So I guess that the AMP approach used to generate the dataset is a different one than used for comparison.

**Robotics Focus:**

4

**Summary Of Paper:**

This paper proposes to use diffusion models for imitation learning of complex multi-skill locomotion policies.

**Summary Of Recommendation:**

See my summary above.

---

### Official Review · Reviewer_wZGM · 2024-07-20

**Originality:** 4
**Technical Quality:** 4
**Clarity Of Presentation:** 3
**Potential Impact:** 3
**Recommendation:** 3
**Confidence:** 3

**Review:**

**Strength:**
- The paper is generally well-written.
- The ability to transition between skills is very important, and to my best knowledge, quite novel.
- Training from both bipedal robot data and quadrupedal robot data is important. The result of making a quadrupedal robot perform bipedal walking is compelling.
- The results on the robustness against different ground conditions is interesting.
- Making real-time inference possible for a relatively large transformer-based diffusion policy.
- Real robot experiments are a plus.

**Weaknesses:**
- The organisation of the sections can be somewhat improved. Quantitative results are a bit few in the main text.
- Some parts can be explained more clearly in the main text, such as RHC framework. Additionally, the data collection for training are not clearly explained in the main text, which I think is quite important.
- There are no quantitative results for bipedal robot experiments
- The names of the rows of the hyperparameters table (Appendix C.3) should match the notation in the paper for readability. For example, I assume “Predict Length” means n-step in the future action prediction?

**Minor notes:**
- I cannot seem to find the final K iteration for diffusion used in the proposed method.
- The bipedal robot experiments are done in simulation as far as I can gather.
- Some citations can be added for better readability; such as AMP in section 6.
- Examples of the goals (commands) given for the shown skill transition would be nice.

**Quality Of The Limitations Section:**

3

**Questions For Rebuttal:**

1. During data collection, how are goals (commands) determined during the collection of offline training datasets? Do the authors determine the goal for the locomotion tasks?

2. To my understanding, the proposed method implicitly learns the “transition” based on the goal. During training, how does the model determine which “goal” belongs to which “skill”? Is it given implicitly by the difference in goals for different skills? That is, are the goal difference between skills large enough such that the proposed method can know which “goal” belongs to which “skill” implicitly?

3. During evaluation, given a same goal, the paper mentioned that two skills might be deployed. How does the policy determine which skill to be the "default" one? As far as I understand, there are no additional information about the external disturbances during training.

4. Following up, during evaluation, I wonder if it is possible to force the robot to perform certain skill? Given a goal, for example, the robot would want to hop. Can the proposed method force the robot to perform bouncing, for example?

5. Robustness to different grounds and terrains is very interesting. As far as I can gather, these information are not provided during training. Are these included in the data collection phase where the policies are trained using different ground and terrains?

6. Somehow hard to quantify, but how smooth are the transition between skills?

7. What is the reason for generating n-step future actions if only the next action will be used? Is it for the stability of training? (An ablation study of n would be nice to have).

8. What is the reason to use $a_{\text{t-h-2:t-2}}$ in line 173 instead of, for example, $a_{\text{t-h-1:t-1}}$ ? Can the authors elaborate?

9. Although not totally related, I wonder about the performance of the proposed method if it is only trained on a single skill dataset? That is, how much that learning a multi-skill policy harm single-skill performance?

**Robotics Focus:**

4

**Summary Of Paper:**

The paper proposes a framework to use a single diffusion-based policy to learn multiple locomotion tasks with pre-collected datasets of individual tasks, ranging from trotting, hopping, walking, pacing and running, across bipedal and quadrupedal robots. The proposed framework is tested on a real-world quadrupedal robot and demonstrates its efficiency in skill transition and robustness against different ground conditions.

**Summary Of Recommendation:**

This paper proposes an interesting approach to tackle the problem of training a multi-skill policy using diffusion models. Overall the writing is clear the empirical results on real-world robots are interesting and I think would be a meaningful contribution to the community. I am leaning towards accepting this paper and I am willing to raise the score higher as long as the concerns listed above are addressed appropriately.

---

### Official Review · Reviewer_8ty4 · 2024-07-21
**A novel diffusion model-based approach for locomotion policies.**

**Originality:** 4
**Technical Quality:** 4
**Clarity Of Presentation:** 4
**Potential Impact:** 4
**Recommendation:** 3
**Confidence:** 4

**Review:**

Strength:
The paper is well-motivated and very well-written and the results are very comprehensive.
The idea of using diffusion for policy representation has been done in manipulation but its application in real-world locomotion is not only novel but also more challenging.
The results section is comprehensive and impressive.



Weakness:
A limitation of such a method can be the length of the trajectory generated. If the task horizons were longer, how does DiffuseLoco scale? The paper uses a relatively small predicted length of 4, an ablation on this on the mujoco humanoid can be really helpful. If this turns out to be a limitation, it should be added to the limitation section.

Considering acceleration was necessary for deployment, was an ablation performed on the size of the model used? How does a smaller model compare to the 6.8M model used in the paper? This ablation can give more insight into how such a model would scale.

Regarding robustness, can you provide a brief explanation as to why the model is robust to different ground conditions? It is not directly obvious why an imitation learning model would be robust to state distribution shifts. Is this robustness a consequence of the diffusion model or high-quality single-skill control policies that generated the data?

**Quality Of The Limitations Section:**

2

**Questions For Rebuttal:**

What is the effect of delayed inputs? Is this a standard implementation detail in locomotion? If yes, please provide a reference. If not, can you provide a simple metric for how much the delayed input helps? Why just a delay of one sample?

The idea of using diffusion for humanoid locomotion from multiple skills appears in [1] as well, albeit they use diffusion to generate a policy model instead of an action sequence, can you provide a brief statement on how DiffuseLoco compares to this? They also achieve strong skill transitioning.

How does DiffuseLoco perform on new tasks/ environments, specifically how many trajectories would be needed to finetune a pre-trained model? The paper mentions that better-quality real-world data can alter performance. How many trajectories of real-world data would be needed to finetune/cotrain the DiffuseLoco model?


[1]: Hegde, Shashank, et al. "Generating behaviorally diverse policies with latent diffusion models." Advances in Neural Information Processing Systems 36 (2023): 7541-7554.

**Robotics Focus:**

4

**Summary Of Paper:**

The paper presents DiffuseLoco, a framework leveraging diffusion models to train multi-skill legged locomotion policies from offline datasets. This framework aims to address the challenges associated with scaling up learning for legged robots by providing a scalable solution that can handle multiple skills within a single policy. DiffuseLoco emphasizes real-time control and robustness in dynamic environments, demonstrating successful zero-shot transfer to real quadrupedal robots.

**Summary Of Recommendation:**

The paper is novel, addresses a challenging question, and has detailed results. Answering a few questions will improve the paper's quality.

---

### Author Rebuttal · Authors · 2024-08-12

Here, we provide two additional plots and videos.
1. Effect of Delayed Input.mp4: This video showcases the effect of delayed inputs. Without delayed inputs, the robot cannot stand up properly when the inference time is 15ms.
2. Multi-skill AMP Baseline in Simulation Env.mp4: This video showcases the performance of multi-skill AMP policy in simulation. The policy is able to walk normally at speed=0.3m/s or 0.7m/s, indicating that the baseline is implemented correctly.
3. Transition Smoothness Joint Pos.pdf: This plot shows the joint positions during transition, indicating interpolation from one skill to the other.
4. Bipedal Robot Quantitative.pdf: This plot shows the velocity tracking performance on bipedal robot transitioning from walking to running.

---

### Decision · Program_Chairs · 2024-09-04

**Decision:**

Accept

**Comment:**

All the reviewers agree that this paper is good, providing an interesting contribution to the literature.
In general, reviewers praise the fact that the authors can show multimodal behavior (bipedal and quadrupedal locomotion)  and that the authors make real-time inference possible for a relatively large transformer-based diffusion policy.

However, one reviewer also states that the only relevant contribution to this work is the codebase. In my opinion (and in contrast to what the reviewers find fundamental), this issue is a pretty strong point against acceptance, as the scope of the contribution for a major conference such as CoRL cannot be limited only to implementation.

I am familiar with this work, and I believe the contributions could go slightly beyond the simple implementation. However, I also believe that the authors should definitively clarify some points and improve other aspects, specifically:
- The claim of smooth transition doesn't seem to be sufficiently substantiated by the experiments. The authors should provide a more in-depth experimentation of this claim. indeed, this claim seems not to be supported by the video material
- The authors should justify the size of the model for a relatively simple task (learning in state space a limited amount of locomotion skills) as a reviewer suggests, this ablation is missing and is particularly crucial for the paper, as the contribution from the scientific point of view is relatively limited.
- Quantitative results for bipedal robot experiments are missing.
- The baseline performance seems to be quite poor. The authors should provide evidence that the baselines are implemented correctly and explain the failure modes more in detail.
- It is fundamental that the authors clarify all the questions of the reviewers, in particular regarding data collection. Looking again quickly at the paper, the authors seem to claim they are using Symmetry Augmented RL. The authors should clarify in detail this. Furthermore,  it would be advisable to add a simple baseline using directly the policies learned with Symmetry Augmented RL to prove that the diffusion model actually can fuse skills better than naively switching these policies.
- If the codebase is the major contribution, I expect this codebase to be available, at least at a preliminary stage, for the rebuttal period. The code base should be executable, and I'll try to check that it can be run following the instructions. Without the codebase, I would not recommend this paper for acceptance.

In general, this paper seems a reasonable contribution, but I would advise the reviewers to carefully check the response of the authors to their points and my comments, and possibly try if it's possible to run the author's code in simulation, as the technical part seems to be prevalent in their work.

===

The reviewers were convinced by the rebuttal and one of the reviewers also was able to test the code, and confirmed it was working.
While I had problems executing the code (due to hardware limitations) I still checked that the instructions were detailed enough and that the overall codebase was reasonable.
There are two main issues remaining: one is that is not clear how to reproduce the gait switching experiments, which should be the main contribution of this paper, and the other is that the codebase still requires a bit of cleanup.
However, given that the authors handled all the issues of the reviewer and provided an open-source link to their code, I believe that the last remaining issues can be easily fixed by the camera-ready version of the paper.